# Dynamics of chikungunya virus transmission in the first year after its introduction in Brazil: A cohort study in an urban community

**Rosângela O. Anjos[1], Moyra M. Portilho[1], Leile Camila Jacob-Nascimento[1], Caroline X. Carvalho[1], Patrícia S. S. Moreira[1], Gielson A. Sacramento[1], Nivison R. R. Nery Junior[1,2], Daiana de Oliveira[1], Jaqueline S. Cruz[1], Cristiane W. Cardoso[3], Hernan D. Argibay[1,2], Kenneth S. Plante[4], Jessica A. Plante[4], Scott C. Weaver[4], Uriel D. Kitron[5], Mitermayer G. Reis[1,6,7], Albert I. Ko[1,7], Federico Costa[1,2,7,8,9], Guilherme S. Ribeiro[1,6]***

1 Instituto Gonçalo Moniz, Fundação Oswaldo Cruz, Salvador, Brazil, 2 Instituto de Saúde Coletiva, Universidade Federal da Bahia, Salvador, Brazil, 3 Secretaria Municipal de Saúde de Salvador, Salvador, Brazil, 4 World Reference Center for Emerging Viruses and Arboviruses, University of Texas Medical Branch, Galveston, Texas, United States of America, 5 Emory University, Atlanta, Georgia, United States of America, 6 Faculdade de Medicina, Universidade Federal da Bahia, Salvador, Brazil, 7 Yale University, New Haven, Connecticut, United States of America, 8 University of Liverpool, Liverpool, United Kingdom, 9 Lancaster University, Lancaster, United Kingdom

* guilherme.ribeiro@fiocruz.br

## Abstract

### Background

The first chikungunya virus (CHIKV) outbreaks during the modern scientific era were identified in the Americas in 2013, reaching high attack rates in Caribbean countries. However, few cohort studies have been performed to characterize the initial dynamics of CHIKV transmission in the New World.

### Methodology/Principal findings

To describe the dynamics of CHIKV transmission shortly after its introduction in Brazil, we performed semi-annual serosurveys in a long-term community-based cohort of 652 participants aged ≥5 years in Salvador, Brazil, between Feb-Apr/2014 and Nov/2016-Feb/2017. CHIKV infections were detected using an IgG ELISA. Cumulative seroprevalence and seroincidence were estimated and spatial aggregation of cases was investigated. The first CHIKV infections were identified between Feb-Apr/2015 and Aug-Nov/2015 (incidence: 10.7%) and continued to be detected at low incidence in subsequent surveys (1.7% from Aug-Nov/2015 to Mar-May/2016 and 1.2% from Mar-May/2016 to Nov/206-Feb/2017). The cumulative seroprevalence in the last survey reached 13.3%. It was higher among those aged 30–44 and 45–59 years (16.1% and 15.6%, respectively), compared to younger (12.4% and 11.7% in <15 and 15–29 years, respectively) or older (10.3% in ≥60 years) age groups, but the differences were not statistically significant. The cumulative seroprevalence was similar between men (14.7%) and women (12.5%). Yet, among those aged 15–29 years, men were more often infected than women (18.1% vs. 7.4%, respectively, P = 0.01), while for those aged 30–44, a non-significant opposite trend was observed (9.3% vs. 19.0%,

**Data Availability Statement:** All relevant data are within the manuscript and its Supporting Information files.

**Funding:** This study was supported by the Brazilian National Council for Scientific and Technological Development (https://www.gov.br/cnpq/pt-br; grants 400830/2013-2, 440891/2016-7, 421522/2016-0, 311365/2021-3 to G.S.R.); the Coordination for the Improvement of Higher Education Personnel, Brazilian Ministry of Education (https://www.gov.br/capes/pt-br; grant 88887.130746/2016-00 to G.S.R.); the Research Support Foundation for the State of Bahia (https://www.fapesb.ba.gov.br/; grant PET0022/2016 to G.S.R.); the US National Institutes of Health (https://www.nih.gov; grants NIAID 5 R01 AI121207, FIC 5 R01 TW009504, FIC 5 R25 TW009338, NIAID 5 U01 AI088752 to A.I.K., and R24 AI AI120942 to S.C.W.); the Wellcome Trust (https://wellcome.org; 102330/Z/13/Z to F.C.); gifts from Sendas Family (to A.I.K.); the Yale School of Public Health; the Oswaldo Cruz Foundation; the Federal University of Bahia; and the Department of Science and Technology, Secretariat of Science, Technology and Strategic Inputs, Brazilian Ministry of Health. The funders had no role in study design, data collection and analysis, decision to publish, or preparation of the manuscript.

**Competing interests:** I have read the journal's policy and the authors of this manuscript have the following competing interests: A.I.K is an expert panel member for Reckitt Global Hygiene Institute and a scientific advisory committee member for Merck-related arboviral research but outside the scope of the manuscript. S.C.W. holds patents for alphavirus vaccine development. Other authors declare no conflict of interest

respectively, P = 0.12). Three spatial clusters of cases were detected in the study site and an increased likelihood of CHIKV infection was detected among participants who resided with someone with CHIKV IgG antibodies.

## Conclusions/Significance

Unlike observations in other settings, the initial spread of CHIKV in this large urban center was limited and focal in certain areas, leaving a high proportion of the population susceptible to further outbreaks. Additional investigations are needed to elucidate the factors driving CHIKV spread dynamics, including understanding differences with respect to dengue and Zika viruses, in order to guide prevention and control strategies for coping with future outbreaks.

### Author summary

The chikungunya virus (CHIKV) was introduced to the Americas in 2013, causing large outbreaks that rapidly affected a substantial portion of the population in several countries. The virus was first detected in Brazil in 2014 and has since spread across the country. However, prospective studies have not been performed to investigate the force of CHIKV transmission shortly after its introduction in Brazil. To fill this gap, we followed 652 participants through a series of six semi-annual serological surveys from Feb/2014 to Feb/2017 in Salvador, Brazil, a city that has been an epicenter of several *Aedes aegypti*-transmitted arbovirus epidemics. As the study started before the detection of CHIKV in Brazil, we were able to estimate the proportion of participants who became infected between each of the surveys by detecting the appearance of CHIKV IgG antibodies. We found that CHIKV transmission was higher between Feb-Apr/2015 and Aug-Nov/2015, when 10.7% of the participants were infected. Transmission was largely focal in space. However, unlike in other American countries, the transmission was limited, with >85% of the participants still susceptible to infection ~1.5 years later. Given the difference in the speed of CHIKV spread among countries, further studies should investigate which factors influence the intensity of CHIKV transmission, aiming to guide prevention and control strategies for coping with future outbreaks.

## Introduction

Chikungunya virus (CHIKV) was first detected in East Africa (in present-day Tanzania) in 1952 [1], but it only emerged as a major public health problem in the current century. In 2004, CHIKV caused an outbreak in Kenya, and from there, it reached the Indian Ocean islands, south and southeast Asia, and even Europe, where it caused small outbreaks in Italy and France [1]. In 2013, the virus established autochthonous transmission in the Caribbean, and in 2014, was first detected in Brazil [2, 3], spreading throughout the country [4,5]. However, the dynamics of CHIKV transmission remain unclear, and several factors may explain the diverse attack rates observed in the different settings where CHIKV was introduced [6–13], such as variations among virus strains and lineages; vector abundance, competence, and control actions; social and behavioral characteristics of the population; research methodology used during investigations (such as the serological test used, the time of survey conduction after

CHIKV introduction, and the criteria for selection of the study population), as well as the multiple interactions among these factors.

Salvador was one of the major Brazilian cities initially affected by CHIKV, with the first outbreak peaking in August 2015. Of note, this outbreak was not promptly recognized because attention was directed to an explosive Zika virus (ZIKV) epidemic in the city a few months earlier [14]. Furthermore, a large community-based seroprevalence survey conducted in Salvador between November 2016 and February 2017 suggested that, compared to ZIKV, CHIKV spread had been limited in the city, as less than 12% of the participants had serological evidence of prior infection [13]. To further describe the dynamics of CHIKV transmission shortly after its introduction in Salvador, we analyzed semi-annual serosurvey data from a long-term community-based cohort study that followed participants from pre- to post-CHIKV emergence in Brazil.

## Methods

### Ethics statement

The study was approved by the Research Ethics Committee of Gonçalo Moniz Institute, Oswaldo Cruz Foundation (CAAE 55904616.4.0000.0040 and 35405320.0.1001.5030), the Brazilian National Commission for Ethics in Research (CAAE 17963519.0.0000.0040) and the Institutional Review Board of Yale University (2000031554). Written informed consent (from participants ≥18 years of age and parents of minors <18 years of age) and written assent (from minors) were obtained from all study participants.

### Study design, setting, and sample

Since 2003, our research group has conducted prospective cohort studies in the community of Pau da Lima, Salvador, Brazil, to investigate the epidemiology of transmissible diseases, such as leptospirosis, Zika, dengue, and COVID-19 [15–20]. The community, a low-income, highly dense, and underserved urban informal settlement, has been extensively characterized [13,15–19,21]. Its features are typical of other low-income areas in Salvador and other large cities in Brazil, especially those in the Northeast region.

During the longitudinal open cohort follow-up performed between Nov/2016-Feb/2017, we visited all households in the study site (comprised of 3 contiguous valleys in Pau da Lima; area of 0.19 Km$^2$) and invited all residents ≥5 years of age who slept ≥3 nights per week in the house to participate in this study about the dynamics of chikungunya transmission. Written informed consent and written assent were obtained to assess whether the cohort participants had developed CHIKV IgG antibodies over the last six consecutive semi-annual cohort follow-up surveys. We also obtained consent to analyze the sociodemographic data collected through interviews with standardized questionnaires during this and the previous surveys. Medical history data were collected in Nov/2016-Feb/2017 through self-reporting of previous presumptive clinical diagnoses of DENV, ZIKV, and CHIKV infection and signs and symptoms common to arbovirus infection, such as fever, arthralgia, myalgia, and rash, at any time after January 2015 since the first cases of chikungunya in Salvador were reported in 2015.

Those who agreed to participate in the Nov/2016-Feb/2017 survey comprised 1,776 (67.2%) of the 2,642 residents in 824 (67.9%) of 1,214 households identified during a census in the area. However, only 654 (36.8%) of them, living in 367 (44.5%) households, had participated in the preceding five consecutive semi-annual cohort follow-up surveys and thus were analyzed in this study (demographic characteristics of the population of Salvador, the neighborhood of Pau da Lima, the study site in Pau da Lima, and the study are shown in S1 Table). The surveys were performed in Feb-Apr/2014, Aug-Nov/2014, Feb-Apr/2015, Aug-Nov/2015, Mar-May/

2016, and Nov/2016-Feb/2017. The period for the first of these surveys was chosen because it preceded the first detection of CHIKV in Brazil.

## Detection of CHIKV infection

Before being tested, all serum samples from the cohort participants were stored at -20°C. Prior CHIKV infection was detected using the CHIKV IgG ELISA from Euroimmun (Luebeck, Germany). The results were interpreted according to manufacturer instructions (CHIKV IgG absorbance/calibrator levels <0.8 were defined as negative, ≥0.8 to <1.1 as indeterminate, and ≥1.1 as positive). Samples with inconclusive results were re-tested and the new results were considered final. Testing was initiated with the sera of participants obtained in the last survey (Nov/2016-Feb/2017), and only those participants with a positive result had the earlier samples (Mar-May/2016) tested. Similarly, those with a positive result in the samples obtained between Mar-May/2016 had the preceding samples (Aug-Nov/2015) tested. We followed this same protocol until all the samples from a given survey were negative. Based on the long-term maintenance of CHIKV IgG antibodies [22,23], we assumed that if a participant had a negative result in a sample obtained in a particular survey, the previously obtained sample would also be negative and so we did not test the preceding samples. This approach substantially reduced the effort and cost compared to testing the entire collection of sera samples obtained during the six surveys; the final number of tested samples was 877.

In addition, CHIKV and Mayaro virus plaque reduction neutralization tests ($PRNT_{50}$) were performed on selected samples to ensure that the presence of CHIKV ELISA antibodies did not represent cross-reactivity between the closely related alphaviruses. Sera were heat inactivated for 30 minutes at 56°C, then serially 1:2 diluted in maintenance media (DMEM supplemented with 2% FBS and 1% penicillin-streptomycin) spanning a range from 1:10 to 1:1,280. The diluted sera was combined with an equal volume of diluted viral stocks (MAYV strain CH or CHIKV strain 181/25) and allowed to incubate for one hour at 37°C with 5% $CO_2$. The mixture was then allowed to infect monolayers of Vero E6 cells for one hour at 37°C with 5% $CO_2$ before being overlaid with 0.4% agarose in 0.8x maintenance media and returned to the incubator for two (MAYV) or three (CHIKV) days. Monolayers were fixed with formalin and stained with crystal violet for plaque count determination. The percent reduction was calculated in relation to control wells lacking sera, which contained on average 33 PFU MAYV and 77 PFU CHIKV.

## Data analysis

During the separation of serum aliquots for testing, we found that 28 participants who had been positive for CHIKV IgG in the last survey (Nov/2016-Feb/2017) did not have all samples available from previous surveys to be tested. To reduce bias in the seroincidence estimates, rather than removing these participants from the analysis, we randomly input a CHIKV IgG status for the participants with missing data in a given survey, weighting the likelihood of a positive result by the measured overall positivity frequency in each of the surveys. For example, since 87 participants were CHIKV IgG positive in the last survey (Nov/2016-Feb/2017), their samples from the preceding survey (Mar-May/2016) should have been tested. However, seven did not have an available sample from this survey; the other 80 participants had their samples tested, and 74 were positive (74/80 = 92.5%; 95% CI: 84.6–96.5%). Based on this frequency of positive samples, we assumed that roughly 92.5% of the seven samples that were not tested should be positive and considered that 6 (85.7% of 7; value within the confidence interval observed to tested samples) should be randomly input as positive and one as negative. Thus, the final number of CHIKV IgG-positive participants in the Mar-May/2016 survey was 80 (74

detected by testing plus six by inputting). The same approach was performed for the other surveys. In addition, we performed sensitivity analyses to compare the findings from this strategy with those from three extreme scenarios: (1) exclusion of the participants with missing data from the surveys in which the sample was not available for testing; (2) considering all the missing data on test result as negative; (3) considering all the missing data on test result as positive (the findings from these analyses are shown in S2 Table).

We used frequencies or means and standard deviations to characterize study participants' sociodemographic and medical histories. These characteristics were compared between those that developed and did not develop a CHIKV infection during follow-up using Poisson regression analysis with robust variance for adjustment for household clustering. A P value <0.05 was used to set a statistically significant difference. As some data may change over time (such as age, years of education, and income), we used the data obtained in the last survey as a standard for these analyses.

The cumulative prevalence of CHIKV IgG antibodies was estimated for each survey by dividing the number of positive participants by the number of participants in the cohort, times 100. The CHIKV seroincidence was estimated for each survey by dividing the number of positive participants in a determined survey by the number of negative participants in the previous survey, times 100. Confidence intervals of 95% (95% CI) for the overall seroprevalences and seroincidences were estimated with adjustment for household clustering using Poisson regression analysis with robust variance. Seroprevalence and seroincidence for each survey were also stratified by sex and age and compared using Poisson regression with robust variance for adjustment for household clustering. For the age-stratified analyses, we used the actual age in each survey.

Data on the reported number of confirmed cases of chikungunya by laboratory or clinical-epidemiological criteria for the city of Salvador and the Community of Pau da Lima between 2014 and 2017 were obtained from the Municipal Health Department, and used to plot a monthly time series graph in order to compare with the cohort seroincidences of CHIKV infection over time. We also georeferenced the households of the cohort participants in the Pau da Lima study site and plotted a map of their distribution according to the participant's CHIKV infection status during follow-up.

To investigate whether the spatial distribution of the households of the CHIKV-infected participants was random, we calculated the global Moran index. Then, we evaluated whether there was a statistically significant aggregation of cases within the area (clusters) by applying the spatial scanning method with a Bernoulli-type distribution using Satscan software [24]. Finally, we verified whether there was an aggregation of cases within households with more than two participants by comparing the frequency of CHIKV infection among participants who resided and did not reside with someone infected with CHIKV. The prevalence ratio and 95% CI for the likelihood of being a case given the occurrence of another case in the household were estimated by Poisson regression with robust variance adjusting for the structure of data dependency within the household. As the number of household members may influence the occurrence of clustering of cases in a household, we also estimated this prevalence ratio adjusted for the number of household members. The analyses to investigate community and household clustering of cases were based on the CHIKV infections detected in Nov/2016-Feb/2017 when no imputation was applied.

## Results

Of the 654 cohort participants, 394 (60.2%) were women, and the median age was 31.3 (SD: 17.7) years. Participants positive for CHIKV IgG were identified in Nov/2016-Feb/2017 (89

participants), Mar-May/2016 (82 participants; 7 who were positive in Nov/2016-Feb/2017 were negative in this survey), and Aug-Nov/2015 (72 participants; 10 who were positive in Mar-May/2016 were negative in this survey). Two participants were CHIKV IgG-positive in the prior survey (Feb-Apr/2015) and in all the earlier ones, including the first, performed in Feb-Apr/2014, when no CHIKV infection had been detected in Brazil.

Therefore, we decided to test earlier samples from these two participants, obtained in surveys performed in Aug-Nov/2013 and Jan-Apr/2013. Again, they were CHIKV IgG-positive. The mean ratio of the ELISA optical density (OD) absorbance value and the calibrator for all the tested samples from these two participants was 1.51 (standard deviation (SD): 0.34; min-max: 1.17–2.28), substantially lower than the mean ratio obtained for all positive samples from the other participants (3.11; SD: 0.68; min-max: 1.11–5.00, overall; 2.54 (SD: 0.53; min-max: 1.44–3.90) in Aug-Nov/2015; 3.14 (SD: 0.54; min-max: 1.19–5.00) in Mar-May/2016; 3.43 (SD: 0.64; min-max: 1.11–4.69) in Nov/2016-Fev/2017) suggesting that the ELISA results from these two participants were likely false-positive. To further address these surprising ELISA results, we performed CHIKV and Mayaro virus $PRNT_{50}$ on the samples of these two participants obtained in Aug-Oct/2014, Feb-Apr/2014, Aug-Nov/2013, and Jan-Apr/2013. All of them were negative for both viruses. Based on the unexpected positivity observed since 2013, the lower OD/calibrator ratios, and the negative results on the $PRNT_{50}$, we classified these two participants as having false-positive CHIKV IgG signals and excluded them from further analysis. Thus, our final cohort comprised 652 participants, of which 87 were considered to have developed CHIKV infection during follow-up.

The sociodemographic characteristics of the 87 participants who developed CHIKV infection were not significantly different from those who did not (Table 1). However, those who developed CHIKV infection most often reported having a medical suspicion of chikungunya, dengue, and Zika, and symptoms compatible with CHIKV infection after 2015, the year of the first CHIKV epidemic in Salvador (P<0.05) (Table 1). The overall frequency in which those infected reported a medical suspicion of chikungunya since 2015 was 9.2%. The occurrence of fever and arthralgia in the same period was reported by 37.2% and 19.5%, respectively.

CHIKV seroincidence was 10.7% (95% CI: 7.6–13.8%) from Feb-Apr/2015 to Aug-Nov/2015, 1.7% (95% CI: 0.6–2.8%) from Aug-Nov/2015 to Mar-May/2016 and 1.2% (95% CI: 0.1–2.3) from Mar-May/2016 to Nov/2016-Feb/2017 (Fig 1 and S3 Table). The seroprevalences of CHIKV IgG were 0.0% in Feb-Apr/2015, followed by 10.7% (95% CI: 7.6–13.8%) in Aug-Nov/2015, 12.3% (95% CI: 9.1–15.5%) in Mar-May/2016, and 13.3% (95% CI: 10.0–16.6%) in Nov/2016-Feb/2017 (Fig 1 and S4 Table). The period with the highest incidence of CHIKV infection in the cohort (from Feb-Apr/2015 to Aug-Nov/2015) coincided with the period of greater notification of chikungunya cases both in Salvador and in the community of Pau da Lima (Fig 1A and 1B).

The 87 CHIKV infections were detected in 64 households located in the three valleys that comprised the study site (Fig 1C), but the spatial distribution of these households was not random, as clustering was observed (Moran index: 0.38, P: 0.001). Spatial scanning identified three statistically significant clusters of infections: the largest comprised 23 cases among 64 participants (Relative Risk: 3.30 in comparison to the overall risk of infection in the area; P < 0.001), the second cluster had 18 cases among 21 participants (Relative Risk: 7.84; P < 0.001) and the third one had 9 cases among 9 participants (Relative Risk: 8.24; P < 0.001). Except for two cases, all the others belonged to a cluster of cases of CHIKV infection occurring between Feb-Apr/2015 and Aug-Nov/2015, suggesting that the clusters were also temporally connected.

Clustering of cases within households was also observed. Of the 87 CHIKV infections detected, almost half (41, 47.1%) occurred in 18 households hosting two or more cases (15

**Table 1. Sociodemographic and self-reported clinical history of study participants according to chikungunya virus (CHIKV) immune status during follow-up.**

| Characteristics | Total participants (N: 652) | CHIKV-uninfected participants (N: 565) | CHIKV-infected participants, according to the follow-up period of infection detection [1] | | | |
|---|---|---|---|---|---|---|
| | | | All infections (N: 87) | Infection detected in Aug-Nov/2015 (N: 70) | Infection detected in Mar-May/2016 (N: 10) | Infections detected in Nov/2016-Feb/2017 (N: 7) |
| | | | Number (%) or mean (SD) | | | |
| **Sociodemographic** | | | | | | |
| Female | 393 (60.3) | 344 (60.9) | 49 (56.3) | 40 (57.1) | 5 (50.0) | 4 (57.1) |
| Mean age, in years | 31.3 (17.6) | 31.2 (17.8) | 32.0 (17.0) | 31.5 (17.1) | 34.4 (17.5) | 33.4 (17.1) |
| Skin color (n = 650) | | | | | | |
| Black/Mixed | 611 (94.0) | 528 (93.8) | 83 (95.4) | 66 (94.3) | 10 (100) | 7 (100) |
| White/Other | 39 (6.0) | 35 (6.2) | 4 (4.6) | 4 (5.7) | 0 | 0 |
| Mean monthly household per capita income, in US $ (n = 651) [2] | 93.5 (92.7) | 96.0 (94.9) | 77.4 (74.9) | 72.3 (72.6) | 101.3 (95.7) | 94.4 (66.8) |
| Years of education [2] | | | | | | |
| None | 25 (3.8) | 20 (3.5) | 5 (5.7) | 3 (4.3) | 1 (10.0) | 1 (14.3) |
| One to five | 229 (35.1) | 195 (34.5) | 34 (39.1) | 30 (42.9) | 3 (30.0) | 1 (14.3) |
| Six to nine | 235 (36.0) | 204 (36.1) | 31 (35.6) | 23 (32.9) | 5 (50.0) | 3 (42.9) |
| At least ten | 163 (25.0) | 146 (25.8) | 17 (19.5) | 14 (20.0) | 1 (10.0) | 2 (28.6) |
| **Self-reported medical history** [2] | | | | | | |
| Medical suspicion of chikungunya [3] | 13 (2.0) | 5 (0.9) | 8 (9.2) | 7 (10.0) | 0 | 1 (14.3) |
| Medical suspicion of dengue [3] | 48 (7.4) | 34 (6.0) | 14 (16.1) | 10 (14.3) | 2 (20.0) | 2 (28.6) |
| Medical suspicion of Zika [3] | 51 (7.8) | 39 (6.9) | 12 (13.8) | 11 (15.7) | 1 (10.0) | 0 |
| Symptoms since January 2015 | | | | | | |
| Fever (n = 651) [3] | 149 (22.9) | 117 (20.7) | 32 (37.2) | 29 (41.4) | 1 (10.0) | 2 (28.6) |
| Arthralgia (n = 648) [3] | 81 (12.5) | 64 (11.4) | 17 (19.5) | 15 (21.7) | 0 | 2 (28.6) |
| Fever and arthralgia (n = 647) [3] | 35 (5.4) | 23 (4.1) | 12 (13.9) | 11 (15.7) | 0 | 1 (14.3) |
| Myalgia | 76 (11.7) | 66 (11.7) | 10 (11.5) | 8 (11.4) | 0 | 2 (28.6) |
| Skin rash (n = 651) | 72 (11.1) | 60 (10.6) | 12 (13.9) | 10 (14.5) | 0 | 2 (28.6) |
| Pruritus (n = 650) | 74 (11.4) | 58 (10.3) | 16 (18.4) | 14 (20.0) | 0 | 2 (28.6) |

[1] No CHIKV infections were detected in the survey performed in Feb-Apr/2015.

[2] Data obtained in the survey performed in Nov/2016-Feb/2017.

[3] P value < 0.05 for the comparison between CHIKV-uninfected and CHIKV-infected participants.

households had 2 cases, 1 had 3 cases, 2 had 4 cases). Furthermore, for the 449 participants who resided in a household with at least one other participant, we found that the frequency of CHIKV infection among those who lived with someone who was IgG-positive was 45.1% (41 of 91), significantly higher than the 5.9% (21 of 358) frequency of CHIKV infection among those living with participants without IgG antibodies (PR: 7.68; 95% CI: 3.99–14.78). This association remained significant when adjusted for the number of residents in the household (PR: 8.37; 95% CI: 4.46–15.72).

Fig 2 and S3 and S4 Tables show seroincidence and seroprevalence of CHIKV infection stratified by age and sex. The cumulative seroprevalence of the last survey (Nov/2016-Feb/2017) was higher among those aged 30–44 and 45–59 years old (16.1% and 15.6%, respectively), followed by <15 years (12.4%), 15–29 years (11.7%) and ≥60 years (10.3%), but the differences were not statistically significant (P = 0.63). Cumulative seroprevalences in Nov/

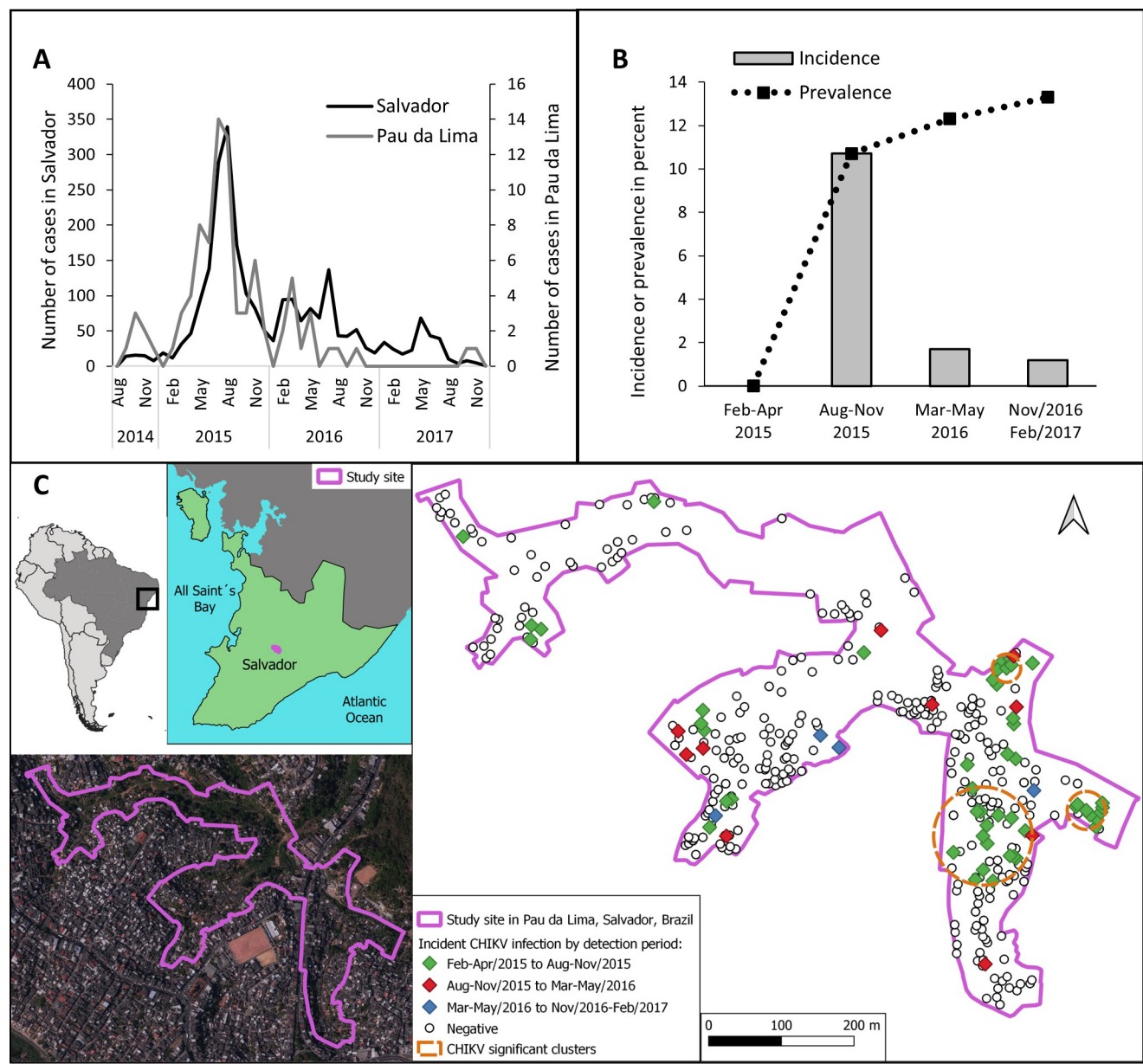

**Fig 1. A) Time series of reported cases suspected of chikungunya (laboratory or clinical-epidemiological diagnosis) in Salvador and the Community of Pau da Lima between 2014–2017 (data provided by the Municipal Health Department). B) Incidence and prevalence of CHIKV infection among the cohort participants in Pau da Lima, Salvador, Brazil, 2015–2017. C) Location of Salvador in Brazil and spatial distribution of households of cohort participants in the Pau da Lima study site according to the participant's CHIKV infection status during follow-up. Clusters of statistically significant cases are shown by the three dashed orange circles.** Note: Incidence was defined based on CHIKV IgG seroconversion between consecutive surveys and prevalence by the cumulative proportion of participants with CHIKV IgG antibodies in each survey. Two subjects who were CHIKV IgG-positive before 2015 were excluded from the study due to suspected CHIKV IgG cross-reactivity. CHIKV IgG immune status was input to 28 participants who were positive in the last survey and did not have serum samples available from previous surveys to test. Source Link to South America map: https://public.opendatasoft.com/explore/dataset/world-administrative-boundaries/export/. Terms of use/license information: https://www.nationalarchives.gov.uk/doc/open-government-licence/version/3/. Source Link to Brazil map and terms of use/license information: https://www.ibge.gov.br/geociencias/organizacao-do-territorio/malhas-territoriais/15774-malhas.html. Source Link to orthoimage from Salvador: http://mapeamento.salvador.ba.gov.br/. Terms of use / license information: http://cartografia.salvador.ba.gov.br/index.php/dados-geoespaciais/baixar-dados-geoespaciais/orientacoes.

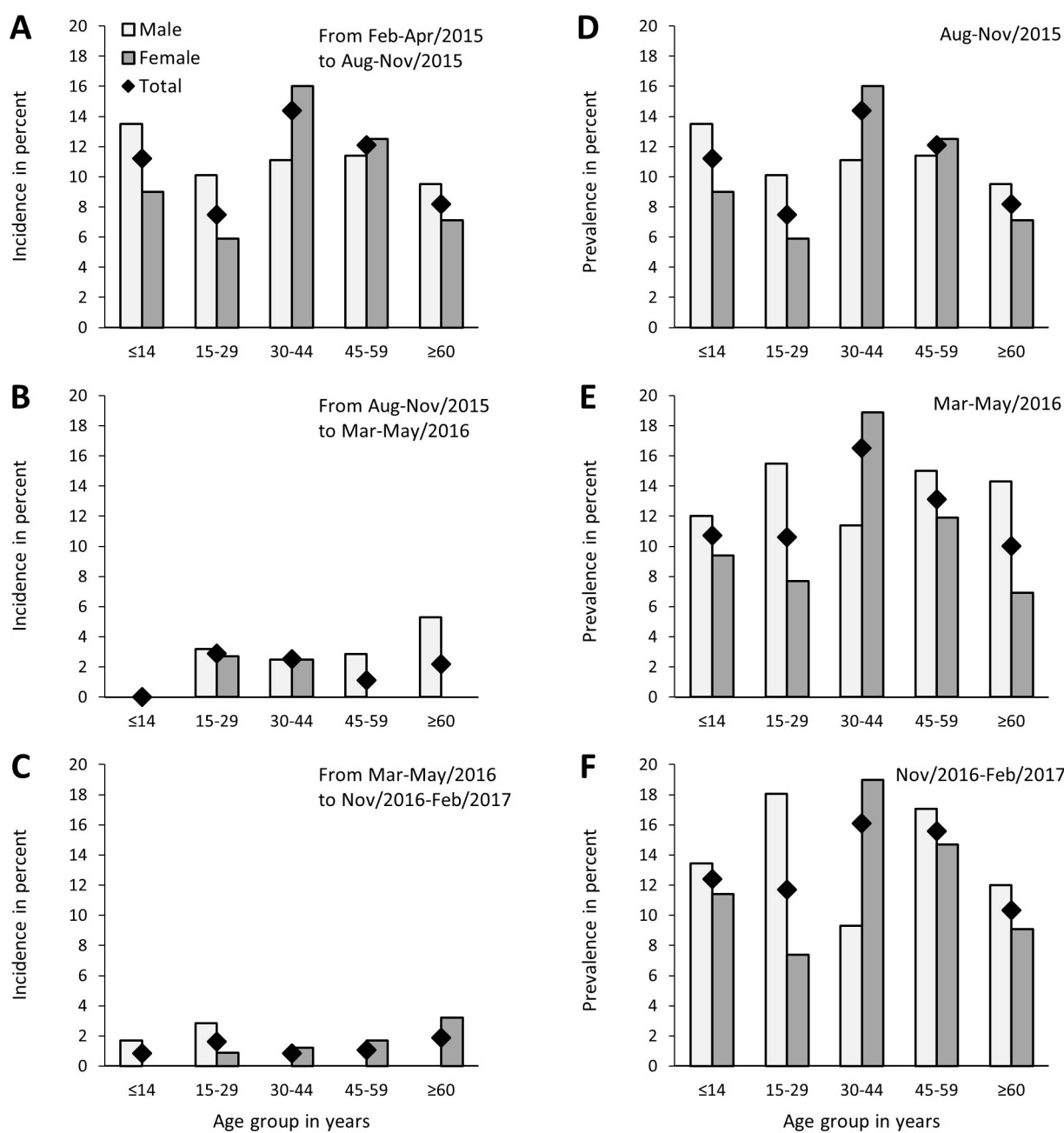

**Fig 2. Incidence and prevalence of CHIKV infection, according to age and sex, Salvador, Brazil, 2015–2017.** Note: Incidence was determined by CHIKV IgG seroconversions between A) Feb-Apr/2015 and Aug-Nov/2015, B) Aug-Nov/2015 and Mar-May/2016, and C) Mar-May/2016 and Nov/2016-Feb/2017. Prevalence was determined by detecting CHIKV IgG in D) Aug-Nov/2016, E) Mar-May/2016, and F) Nov/2016-Feb/2017. CHIKV IgG immune status was input to 28 participants who were positive in the last survey and did not have serum samples available from previous surveys to test.

2016-Feb/2017 were similar for men and women (14.7% vs. 12.5%, respectively, P = 0.38), but among those aged 15–29 years, the seroprevalence was higher in men than women (18.1% vs. 7.4%, respectively, PR: 2.45, 95% CI: 1.21–4.95, P = 0.01). A non-significant difference in the opposite direction was observed for those aged 30–44 years (9.3% vs. 19.0%, respectively, PR: 0.49, 95% CI: 0.18–1.30, P = 0.15). The low seroincidences precluded testing for an association of age and sex with the risk of CHIKV infection between each survey.

## Discussion

Most knowledge on the early dynamics of CHIKV transmission in the Americas has been derived from passive case-patient reporting [3,25–29] or cross-sectional serological surveys [6–9,11–13,26,30–32]. Passive national surveillance systems based on mandatory case reporting have low sensitivity because patients misdiagnosed as having another disease are not reported, and those less symptomatic may not seek health care [33–35]; the former is especially important for CHIKV infection, which is easily confused with dengue or even Zika [36]. On the other hand, cross-sectional surveys may not distinguish infections that occurred during the outbreak from those that happened afterwards, when endemic transmission ensued. Because we had an ongoing community-based cohort study in Brazil, which underwent dense sampling with semi-annual serosurveys before and after CHIKV was first detected in the country, we had a unique opportunity to examine the initial dynamics of virus spread through a series of six serological surveys performed between 2014 and 2016.

Our study confirmed that the first wave of CHIKV transmission in Salvador occurred in the second half of 2015 [14,36]. It also indicated that this initial outbreak was limited, during which only 10.7% of the study participants were exposed by Aug-Nov/2015. Furthermore, during 2016, after the CHIKV outbreak peak, endemic transmission remained low, with ~3% of those still susceptible becoming infected. By Nov/2016-Feb/2017, ~1.0–1.5 years after the initial CHIKV transmission in Salvador, the cumulative CHIKV seroprevalence in our study population was only 13.3%, similar to the seroprevalence of 11.8% that we found in a large cross-sectional study performed in this same community in Nov/2016-Feb/2017 [13]. In addition, we identified clustering of CHIKV infections in certain areas of the study site and within households. Spatial and household aggregation has been described for CHIKV and other human-amplified, *Aedes aegypti*-borne arboviral diseases at different scales in several areas around the globe [37]. It is noteworthy that the clusters in our study were spatially limited, despite the high density of the population and socio-environmental similarities within the community, which could have facilitated the expansion of CHIKV transmission. Further investigations are needed to elucidate which spatial factors serve as drivers or constrictors of CHIKV spread, and to determine the role of focal spread of CHIKV during epidemic and endemic transmission periods.

The absence of other cohort studies to investigate the early CHIKV transmission in Brazil limits national-level comparisons of the rate of CHIKV spread over time. However, a cohort study initiated before the introduction of CHIKV in Managua, Nicaragua, which followed children 2 to 14 years of age through annual serological surveys, found a small CHIKV epidemic wave between 2014 and 2015 and a second larger one between 2015 and 2016 (incidence rates of 6.1 and 21.8 infections per 100 person-years, respectively) [38]. Although both the Nicaraguan study and ours observed low attack rates in the first year after CHIKV introduction, virus transmission significantly increased during the following year in Managua, which was a trend we did not observe in our cohort in Brazil.

Furthermore, cross-sectional surveys conducted in the Americas roughly one year after CHIKV introduction found higher attack rates, ranging between 20.0% in a rural area of northeast Brazil [6] to 20.7% in the Saint Martin Island [39], 23.5% in Puerto Rico [8], 27.0% in Ecuador [7], 31.0% in the United States Virgin Islands [31], 32.8% in Nicaragua [10], 41.9% in Martinique [9], 48.1% in Guadeloupe [9], 57.9% in Haiti [11], and 45.7% and 57.1% in two urban areas of northeastern Brazil [12]. These findings suggest that the pattern of CHIKV transmission among naïve populations is diverse and not easily predictable.

We can speculate on possible reasons for the contained transmission of CHIKV seen in Pau da Lima. The co-circulation of other arboviruses, particularly the ZIKV, which caused an

explosive epidemic shortly before the peak of CHIKV transmission in Salvador in 2015 [40,41], may have created an ecological competition between these viruses for the same vector, which might (at least theoretically) have contributed to restrict the spread of CHIKV. As a comparison, while we detected that 10.7% of the participants of this study had been exposed to CHIKV by August-November/2015, another survey we conducted in the same period and community found that 73% of participants had been exposed to ZIKV [17].

However, experiments with cultured cells suggest that ZIKV and CHIKV can co-infect and co-replicate without ZIKV interference with CHIKV replication kinetics in either *Ae. albopictus* or *Ae. aegypti* cells [42]. Also, even during major epidemics, infection rates of these vectors rarely exceed 1%, with dual infections even more infrequent. Furthermore, the co-infection of *Ae. aegypti* by CHIKV and ZIKV was shown not to substantially reduce the vector's capacity to be infected or transmit either virus [42–45]; but in the real world, a vector is more likely to experience a superinfection (when different viruses sequentially infect it) than a co-infection [46]. Sequential flavivirus infections can result in viral interference (when a primary virus inhibits infection from a secondary virus) in mosquito cell lines [47]. Whether this may also happen in superinfections between ZIKV and CHIKV (or between CHIKV and another mosquito microbiota) remains less clear, but experimental investigations suggest that CHIKV and ZIKV do not compete during sequential infection of *Ae. aegypti* mosquitoes [48].

Another possible explanation for our cohort's low CHIKV attack rate is a potential strengthening of vector control and personal protection actions in response to the ZIKV epidemic in Salvador, which could have reduced the risk of CHIKV transmission. Alternatively, the ZIKV outbreak may have started in a more favorable seasonal period for vector populations than the CHIKV outbreak. However, chikungunya cases were occurring in Salvador before ZIKV transmission was detected. Moreover, a citywide laboratory-based surveillance study we conducted in Salvador found that CHIKV transmission had been gradually increasing since the first weeks of 2015, being outweighed by the explosive nature of the ZIKV outbreak [14]. Thus, it is unlikely that seasonal fluctuations in the mosquito population or vector control actions have a differential impact on the transmission of these two viruses, given that their transmission timeframes overlap. Further investigations are needed to elucidate why ZIKV and CHIKV had different dissemination patterns in the same naïve community and to understand why CHIKV had diverse dynamics of spread in the Americas.

Although not significantly different, we found that the highest CHIKV transmission tended to occur among individuals aged 30–59 years compared to younger or older ages. Some studies have also observed the same trend [10,32], while others have shown a higher frequency of CHIKV antibodies among children [49] or the elderly [9,39,50]. Interestingly, we found that the overall cumulative risk of infection was significantly higher for men among those aged 15–29 years and non-significantly higher for women among those aged 30–44 years. Differences in sex- and age-related behavior or body composition may have affected the risk of CHIKV infection by increasing or reducing the likelihood of human-vector interactions. It has been shown that the probability of an individual being bitten by *Ae. aegypti* mosquitoes at home is directly proportional to factors that are correlated with age and sex, such as length of stay at home and body surface area [51] (e.g., adults and men tend to have greater surface area than children and women; children, elderly, and women may spend more time at home than adults and men).

The frequency of symptoms commonly observed during acute CHIKV infection was low among those who developed CHIKV IgG antibodies during follow-up. In addition, less than 10% of infected participants had a clinical suspicion of chikungunya. These findings may indicate that most of the CHIKV infections in our cohort were inapparent. While it is widely accepted that <15% of CHIKV infections are asymptomatic [1], some studies have found

frequencies of inapparent infections >60% in regions where the Asian CHIKV genotype–which is considered less virulent than others [52]–caused outbreaks [32,53].

In Brazil, both the Asian and the East-Central-South-African (ECSA) CHIKV genotypes have circulated since 2014 [2, 3], but only the ECSA genotype has been detected in northeastern Brazil, where Salvador is located [2,54,55]. Although previous cross-sectional studies in this region have also found high frequencies of inapparent infections (ranging from 45.8% to 84.7% [6,12,13]), prior medical history data in these studies, as well as in the current one, were obtained between one and one and a half years post-infection. Furthermore, during the first peak of CHIKV transmission in 2015, clinicians were not fully aware of the CHIKV outbreak, and CHIKV-infected participants were more commonly suspected to have contracted dengue or Zika rather than chikungunya [36]. Therefore, recall bias is also a possible cause for the low rate of symptomatic infection. Prospective cohort studies with active surveillance for arboviral symptoms among the cohort members are needed to assess the frequency of symptomatic CHIKV infections caused by the ECSA genotype.

Our study has other limitations. It was conducted in only one community in Salvador, and the temporal pattern of CHIKV propagation in the city may not be unique. For example, in May 2017, a focal outbreak of CHIKV was detected in Salvador, mainly affecting a single street, which suggests that localized transmission and spatial heterogeneity may occur in the spread of CHIKV in a large urban city [55]. However, consistent with current findings, our previous investigations and citywide data on reported chikungunya cases also found that the initial peak of CHIKV transmission in Salvador occurred after mid-2015 [14,36] and was followed by a period of low CHIKV transmission that lasted until 2019 [56,57].

Furthermore, given the peak period of the outbreak, we cannot rule out that some participants with a negative CHIKV IgG test result in the Aug-Nov/2015 survey had very recently been infected but had not developed IgG antibodies, underestimating the attack rate for the period between Feb-Apr/2015 and Aug-Nov/2015. On the other hand, such infections would be detected in the following period, leading to an overestimation of infections between Aug-Nov/2015 and Mar-May/2016. In addition, some participants did not have one or more serum samples available for testing, and we imputed a value for the presence or absence of CHIKV IgG antibodies. This may result in underestimating or overestimating the incidences and prevalences measured in the surveys performed between Feb-Apr/2015 and Mar-May/2016, but not for the last survey performed in Nov/2016-Feb/2017 when all the participant samples were available for testing. Furthermore, the proportion of participants that underwent imputation was <5%, and we used a random imputation based on the expected probability of the test result to obtain a minimally biased value.

Finally, like all serological assays, the one we employ is not error-free. A meta-analysis study showed that the used test has a sensitivity of 95.5% and a specificity of 91.5% [58], which indicates that cross-reactions may be a limitation of the test. However, cross-reactions typically occur when the test is used on serum samples obtained from individuals previously exposed to another virus in the same family. Currently, there is no evidence that other alphaviruses are transmitted in Salvador, although the closely related Mayaro virus occurs in other parts of Brazil, particularly in the Amazon region [59].

We previously evaluated the agreement between the assay used and the plaque-reduction neutralization test (PRNT) for CHIKV using sera from 60 Pau da Lima residents. We found no IgG-positive sample among the PRNT-negative ones [13]. Yet, during this study, we found two participants with low levels of CHIKV IgG antibodies in samples collected more than a year before the first identification of CHIKV in Brazil. Although these antibodies likely represent a nonspecific signal, we could not determine whether these individuals were exposed to a

previous CHIKV or other alphavirus infection while traveling to an area where these viruses circulate.

In summary, our study reports the seroincidence of anti-CHIKV antibodies during the first CHIKV epidemic in Salvador, Brazil. It confirms findings from previous investigations suggesting that the initial spread of CHIKV in this large urban center was limited, both in magnitude and in time. It also supports earlier findings showing that CHIKV transmission can be highly focal [55], clustering within households and in certain community areas. The low incidence and minimal herd immunity levels less than three years after the CHIKV introduction place 85% of the population at risk for new outbreaks. This transmission pattern contrasts with most American countries, where larger epidemics were recorded following the CHIKV introduction, resulting in greater herd immunity. Additional investigations are needed to elucidate better which factors drive CHIKV spread dynamics, as this knowledge can help guide prevention and control strategies for mitigating future CHIKV outbreaks.

## Supporting information

**S1 Table. Demographic characteristics of the population of Salvador, the neighborhood of Pau da Lima, the study site in Pau da Lima, the sample of participants in the survey conducted in Nov/2016-Feb/2017 survey, and the cohort members who participated in all surveys between Feb-Apr/2014 and Nov/2016-Feb/2017.**
(DOCX)

**S2 Table. Sensitivity analyses on the extreme situations for missing data (excluding the samples with missing results, considering the samples with missing results as all positive or as all negative) in comparison to those derived from the participants with complete data and the random imputation method used, weighted by the likelihood of a positive result for the sample with missing data.**
(DOCX)

**S3 Table. Incidence of chikungunya virus infection according to age and sex, Salvador, Brazil, 2015–2017.**
(DOCX)

**S4 Table. Prevalence of chikungunya virus infection according to age and sex, Salvador, Brazil, 2015–2017.**
(DOCX)

**S1 Appendix. Database and codebook for the article "Dynamics of chikungunya virus transmission in the first year after its introduction in Brazil: A cohort study in an urban community".**
(XLS)

## Acknowledgments

We want to thank the technical staff who participated in study data collection, sample processing, data management, and regulatory and administrative matters; the community of Pau da Lima for their support for the investigation; and most of all, the study participants.

## Author Contributions

**Conceptualization:** Rosângela O. Anjos, Scott C. Weaver, Uriel D. Kitron, Mitermayer G. Reis, Albert I. Ko, Federico Costa, Guilherme S. Ribeiro.

**Data curation:** Rosângela O. Anjos, Nivison R. R. Nery Junior, Albert I. Ko, Federico Costa, Guilherme S. Ribeiro.

**Formal analysis:** Rosângela O. Anjos, Hernan D. Argibay.

**Funding acquisition:** Albert I. Ko, Federico Costa.

**Investigation:** Rosângela O. Anjos, Moyra M. Portilho, Leile Camila Jacob-Nascimento, Caroline X. Carvalho, Patrícia S. S. Moreira, Gielson A. Sacramento, Nivison R. R. Nery Junior, Daiana de Oliveira, Jaqueline S. Cruz, Cristiane W. Cardoso, Kenneth S. Plante, Jessica A. Plante.

**Methodology:** Rosângela O. Anjos, Hernan D. Argibay, Scott C. Weaver, Uriel D. Kitron, Mitermayer G. Reis, Albert I. Ko, Federico Costa, Guilherme S. Ribeiro.

**Project administration:** Rosângela O. Anjos, Moyra M. Portilho, Gielson A. Sacramento, Nivison R. R. Nery Junior, Albert I. Ko, Federico Costa, Guilherme S. Ribeiro.

**Resources:** Cristiane W. Cardoso, Scott C. Weaver, Albert I. Ko, Federico Costa, Guilherme S. Ribeiro.

**Supervision:** Albert I. Ko, Federico Costa, Guilherme S. Ribeiro.

**Validation:** Rosângela O. Anjos, Caroline X. Carvalho, Patrícia S. S. Moreira.

**Visualization:** Rosângela O. Anjos, Hernan D. Argibay.

**Writing – original draft:** Rosângela O. Anjos, Guilherme S. Ribeiro.

**Writing – review & editing:** Rosângela O. Anjos, Moyra M. Portilho, Leile Camila Jacob-Nascimento, Caroline X. Carvalho, Patrícia S. S. Moreira, Gielson A. Sacramento, Nivison R. R. Nery Junior, Daiana de Oliveira, Jaqueline S. Cruz, Cristiane W. Cardoso, Hernan D. Argibay, Kenneth S. Plante, Jessica A. Plante, Scott C. Weaver, Uriel D. Kitron, Mitermayer G. Reis, Albert I. Ko, Federico Costa, Guilherme S. Ribeiro.

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
