## [Decision Letter · Decision Letter 0]

7 Jul 2023

Dear Dr.. Ribeiro,

Thank you very much for submitting your manuscript "Dynamics of chikungunya virus transmission in the first year after its introduction in Brazil: A cohort study in an urban community" for consideration at PLOS Neglected Tropical Diseases. As with all papers reviewed by the journal, your manuscript was reviewed by members of the editorial board and by several independent reviewers. In light of the reviews (below this email), we would like to invite the resubmission of a significantly-revised version that takes into account the reviewers' comments. 

Overall, despite the data being quite dated at this point, I think this type of longitudinal data is valuable for understanding novel arboviral introductions into naive populations. As the authors note, it is interesting that CHIKV did not take off past the initial introduction in Salvador, Brazil, as it did in some regions. The data presented here are also consistent with what the authors had found in a previous study in the same region. Also, I appreciate that the authors provided the data in the supplemental material for future analysis.

The reviewers had differing assessments of this manuscript, but both raised some comments to address. I've also added a few additional comments to consider below. 

- I agree with Reviewer 2 that it would be helpful to have a cursory description of the cohort in the Methods (e.g., recruitment approach, number of households, how/when symptom data were collected, etc). I realize that this info can largely be found in other publications, but it's helpful to have it here in sufficient detail for a reader without chasing down past publications. How large was the cohort area, and is it representative of the greater urban area?

- Reviewer 1 requests IgM data. While I agree this data would be nice to have, this is completely optional - if you have it, great. If not, I would not expect it to be generated for this manuscript.

- Both reviewers were interested in more context for these results, particularly regarding reported cases in Salvador or the region of Brazil over the course of the study. If this data is available, consider adding in a figure with the time series of cases (e.g. by month), either to the main text or as a supplemental figure. However, I think Table 1 is valuable and disagree with the recommendation to modify it. Additionally, one of the reviewers requested a geographic distribution of cases - I agree this would be useful, but I would consider it optional, since I assume the MoH data would not necessarily be provided with addresses or lat/long. 

- Lines 120-121: It would be helpful to include a bit more information on the use of the Eurimmune ELISA. How were positives determined, for example?

- Line 135: Having 28 out of 89 (or 87) is a large number of missing samples - this should be noted as a limitation in the Discussion. Does the random imputation potentially bias the data in some way? What seroprevalence number was used for the estimates? That should be listed here. As one of the reviewers indicated, this approach was a bit difficult to follow. 

- Line 170: While OD of 0.51 is less than the mean, how does it compare to baseline and the distribution of other positive samples? It would be helpful to see the distribution of ODs in the supplemental material, or some sort of summary of the data in the Results (e.g., cutoff value was x, positive samples ranged from x to y, with a median of z, something along those lines). This goes back to the question about methods and the positive/negative cut-off value.

- Figure 1: I recommend an asterisk that says that 2 individuals that were positive earlier in the study were removed because of suspected cross-reaction. Also add in a note that results from 28 individuals were imputed.

We cannot make any decision about publication until we have seen the revised manuscript and your response to the reviewers' comments. Your revised manuscript is also likely to be sent to reviewers for further evaluation.

Sincerely,

Abdallah M. Samy, PhD

Section Editor

Abdallah Samy

Section Editor

Overall, despite the data being quite dated at this point, I think this type of longitudinal data is valuable for understanding novel arboviral introductions into naive populations. As the authors note, it is interesting that CHIKV did not take off past the initial introduction in Salvador, Brazil, as it did in some regions. The data presented here are also consistent with what the authors had found in a previous study in the same region. Also, I appreciate that the authors provided the data in the supplemental material for future analysis.

The reviewers had differing assessments of this manuscript, but both raised some comments to address. I've also added a few additional comments to consider below. 

- I agree with Reviewer 2 that it would be helpful to have a cursory description of the cohort in the Methods (e.g., recruitment approach, number of households, how/when symptom data were collected, etc). I realize that this info can largely be found in other publications, but it's helpful to have it here in sufficient detail for a reader without chasing down past publications. How large was the cohort area, and is it representative of the greater urban area?

- Reviewer 1 requests IgM data. While I agree this data would be nice to have, this is completely optional - if you have it, great. If not, I would not expect it to be generated for this manuscript.

- Both reviewers were interested in more context for these results, particularly regarding reported cases in Salvador or the region of Brazil over the course of the study. If this data is available, consider adding in a figure with the time series of cases (e.g. by month), either to the main text or as a supplemental figure. However, I think Table 1 is valuable and disagree with the recommendation to modify it. Additionally, one of the reviewers requested a geographic distribution of cases - I agree this would be useful, but I would consider it optional, since I assume the MoH data would not necessarily be provided with addresses or lat/long. 

- Lines 120-121: It would be helpful to include a bit more information on the use of the Eurimmune ELISA. How were positives determined, for example?

- Line 135: Having 28 out of 89 (or 87) is a large number of missing samples - this should be noted as a limitation in the Discussion. Does the random imputation potentially bias the data in some way? What seroprevalence number was used for the estimates? That should be listed here. As one of the reviewers indicated, this approach was a bit difficult to follow. 

- Line 170: While OD of 0.51 is less than the mean, how does it compare to baseline and the distribution of other positive samples? It would be helpful to see the distribution of ODs in the supplemental material, or some sort of summary of the data in the Results (e.g., cutoff value was x, positive samples ranged from x to y, with a median of z, something along those lines). This goes back to the question about methods and the positive/negative cut-off value.

- Figure 1: I recommend an asterisk that says that 2 individuals that were positive earlier in the study were removed because of suspected cross-reaction. Also add in a note that results from 28 individuals were imputed.

Reviewer's Responses to Questions

**Key Review Criteria Required for Acceptance?**

**Methods**

-Are the objectives of the study clearly articulated with a clear testable hypothesis stated?

-Is the study design appropriate to address the stated objectives?

-Is the population clearly described and appropriate for the hypothesis being tested?

-Is the sample size sufficient to ensure adequate power to address the hypothesis being tested?

-Were correct statistical analysis used to support conclusions?

-Are there concerns about ethical or regulatory requirements being met?

Reviewer #1: Objectives clearly articulated, study design appropriate, population clearly described and appropriate. Sample size sufficient and correct statistical analysis. No ethical concerns.

Reviewer #2: The authors aimed at describe the dynamics of CHIKV transmission shortly after its relatively recent introduction in Brazil, using semi-annual sero-surveys by following up a community-based cohort of 652 participants aged ≥5 years in Salvador, Brazil, between Feb-Apr/2014 and Nov/2016-Feb/2017.

The first question is related to the so-called community-based cohort of 652 participants. Can the authors elaborate further? How many households and what was the median size of a household? Were all the members of the household recruited and followed up? What is the approx. size of the geographical area of recruitment? How were the participants approached and what was the strategy for their recruitment? Is the sample population representative for Salvador - given a very low figure for CHIKV seroprevalence?

I do not see a point for such a detailed Table 1. Rather, I would like to see summary statistics of the sample beside Salvador’s population statistics. Also, there should be surveillance figures, more time-granulated, for example, weekly or monthly figures in the population. How do the figures in the cohort 

I am not entirely sure if I understand the method below. Why not taking extreme situations for missing data and see how sensitive the estimates might be compared to those derived on complete data.

“To reduce bias in the seroincidence estimates, rather than removing these participants from the analysis, we randomly input a CHIKV IgG status for the participants with missing data in a given survey, weighting the likelihood of a positive result by the measured seroprevalence in each of the surveys.”

I am not sure how the authors did what they describe below – empirical methods for the cumulative prevalence and sero-incidence, yet adjusting for household clustering.

“The cumulative prevalence of CHIKV IgG antibodies was estimated for each survey by dividing the number of positive participants by the number of participants in the cohort, times 100. The CHIKV seroincidence was estimated for each survey by dividing the number of positive participants in a determined survey by the number of negative participants in the previous survey, times 100. Confidence intervals of 95% (95% CI) for the overall seroprevalences and seroincidences were estimated with adjustment for household clustering.”

The presentation of the data is just empirical, the panels in the figure 2 are just arithmetic quantities. The interpretation as in “men were more often infected than women…” is not appropriate, perhaps more likely.

I am not entirely sure about the presentation and importance of these data. The researchers hold individual longitudinal data – yet they investigate it as aggregated data.

**Results**

-Does the analysis presented match the analysis plan?

-Are the results clearly and completely presented?

-Are the figures (Tables, Images) of sufficient quality for clarity?

Reviewer #1: Results clearly presented and the figures sufficient quality for clarity.

Reviewer #2: I do not feel that the presented analyses match the analysis presented plan. Quite contrary, the results presentation is empirical and I do not quite understand how the authors applied their Poisson regression and how the manuscript benefited from it.

**Conclusions**

-Are the conclusions supported by the data presented?

-Are the limitations of analysis clearly described?

-Do the authors discuss how these data can be helpful to advance our understanding of the topic under study?

-Is public health relevance addressed?

Reviewer #1: Conclusions supported by data. Limitations as noted in my major and minor comments attached. Good discussion and public health relevance addressed.

Reviewer #2: I do not understand what is the population the researchers aim at making inference about using their sample data - namely what the sample data represent.

PLOS authors have the option to publish the peer review history of their article (what does this mean?). If published, this will include your full peer review and any attached files.

Reviewer #1: No

Reviewer #2: Yes: Dr IRINA CHIS STER

**Editorial and Data Presentation Modifications?**

Reviewer #1: The manuscript entitled, “Dynamics of chikungunya virus transmission in the first year after its

introduction in Brazil: A cohort study in an urban community” PNTD-D-23-00347, Rosângela O. Anjos et al was reviewed.

Thank you for the opportunity to review this manuscript. The authors describe the dynamics of CHIKV transmission shortly after its introduction in Brazil, by performing semi-annual serosurveys in a long-term community-based cohort of 652 participants aged ≥5 years in Salvador, Brazil, between Feb-Apr/2014 and Nov/2016-Feb/2017. CHIKV infections were detected using CHIKV IgG ELISA. The cumulative seroprevalence in the last survey reached 13.3%. The authors describe a seroprevalence higher among those aged 30-44 and 45-59 years (16.1% and 15.6%, respectively), compared to younger (12.4% and 11.7% in <15 and 15-29 years, respectively) or older (10.3% in ≥60 years) age groups. The cumulative seroprevalence was similar between men (14.7%) and women (12.5%); among those aged 15-29 years, men were more often infected than women (18.1% vs. 7.4%, respectively, P = 0.01), while for those aged 30-44, a non-significant opposite trend was observed (9.3% vs. 19.0%, respectively, P = 0.12). The authors concluded that the initial spread of CHIKV in this large urban center was limited, leaving a high proportion of the population susceptible to further outbreaks. 

The manuscript describes a continued important arboviral disease, CHIKV, with regional large outbreaks of disease ie Paraguay currently. The manuscript is well-written and of a topic of interest to the readers of PLOS NTD. 

Major Comments:

1. This is a descriptive study but as a cohort study offers potential valuable insight into CHIKV disease and transmission. Important questions that could be answered relevant to the broader understanding of CHIKV is the burden of disease and symptom complex severity in children, changing mortality and morbidity, burden of chronic illness associated with acute infection, and risk of household contacts. To that end additional information and data from this study would provide insight into important questions about CHIKV:

a. Symptoms reported in table one, when obtained after infection?

b. Symptoms broken down by age in particular the pediatric population.

c. Deaths reported from CHIKV or hospitalizations.

d. An estimate of the proportion of subclinical infections in total and by age.

e. An estimate of household clustering of cases and risk of infection.

2. Of value to mention the circulating strain of CHIKV during that time period in Salvador, I’m assuming its ECSA, or even better an isolate from the cohort study and strain.

Minor Comments:

1. CHIKV is a reportable disease in Brazil, it would be of value to provide context into what is being reported outside of the study area. A map of the study area within a larger map of Salvador will assist the readers in the context of the study geographic area as well as reported CHIKV for Salvador to the MoH as an epidemic curve over time during the serosurveys would be useful.

2. Geographic distribution of cases over time on the map would be useful to the spatial spread.

3. Figure 2, would be useful to have an additional sub-figure showing total incidence and prevalence of CHIKV infection, according to age and sex by year.

4. The data stands by itself, but the authors could consider adding IgM to the testing (understanding the budget limitations of testing etc) as the duration of IgM following acute infection is 3 to 4 months and would provide more fidelity on the time of infection and potentially capture cases before IgG became positive ie the last serological capture point.

**Summary and General Comments**

Reviewer #1: A well written manuscript providing additional information on the transmission of CHIKV in one community in Brazil and adding to the literature of the burden of disease and temporal transmission of this virus. My comments would increase the breadth of knowledge gained from this study.
---

## [Decision Letter · Decision Letter 1]

9 Nov 2023

Dear Dr.. Ribeiro,

Thank you very much for submitting your manuscript "Dynamics of chikungunya virus transmission in the first year after its introduction in Brazil: A cohort study in an urban community" for consideration at PLOS Neglected Tropical Diseases. As with all papers reviewed by the journal, your manuscript was reviewed by members of the editorial board and by several independent reviewers. The reviewers appreciated the attention to an important topic. Based on the reviews, we are likely to accept this manuscript for publication, providing that you modify the manuscript according to the review recommendations. 

The authors have addressed the major concerns of the reviewers. While some reasonable concerns remain from one of the reviewers, I consider these to be largely related to data interpretation, and I believe the authors should have some leeway to interpret their findings.

Sincerely,

Brett M. Forshey

Academic Editor

Abdallah Samy

Section Editor

The authors have addressed the major concerns of the reviewers. While some reasonable concerns remain from one of the reviewers, I consider these to be largely related to data interpretation, and I believe the authors should have some leeway to interpret their findings.

Reviewer's Responses to Questions

**Key Review Criteria Required for Acceptance?**

**Methods**

-Are the objectives of the study clearly articulated with a clear testable hypothesis stated?

-Is the study design appropriate to address the stated objectives?

-Is the population clearly described and appropriate for the hypothesis being tested?

-Is the sample size sufficient to ensure adequate power to address the hypothesis being tested?

-Were correct statistical analysis used to support conclusions?

-Are there concerns about ethical or regulatory requirements being met?

Reviewer #1: Authors have addressed the reviewers' comments adequately on the methodology of this manuscript.

Reviewer #2: The objectives of the study are reasonably narrated. The longitudinal aspect of the study aims at answering researchers' questions. The description of the population improved although it is not entirely clear whether the sample represents the population. But efforts have been made in this sense. This is an observational study aiming at describing the epidemiology of an infective agent hence an a priori sample size is not necessary.

The statistical methodology description has improved. I have no concerns about ethical or regulatory requirements.

I am not entirely sure about the title of the communication, namely transmission. I find it little relevant as the authors did not derive key dynamics elements such a force of infection which would make possible to calculate some periodic R0. Their entire analysis in merely descriptive. Also, the incidence estimates are not robust, perhaps prevalence would suffice to describe the data. It is reasonable to assume that CHIKV induces long life immunity hence the prevalence of past exposure may be a better epidemiological measure to refer to and describe the disease.

**Results**

-Does the analysis presented match the analysis plan?

-Are the results clearly and completely presented?

-Are the figures (Tables, Images) of sufficient quality for clarity?

Reviewer #1: Authors have addressed the reviewers' comments adequately on the analysis and results of this manuscript.

Reviewer #2: The analysis generally matched the plan. The graphical representation of the data are also more informative in this version and efforts are appreciated in this sense. I am not entirely sure about the incidence estimates. Due to low numbers, Table S3 is full of altered 95%CIs and I am not sure of their relevance. Best to present just the numbers and no estimation at all.

line 371: "...endemic transmission remained low, with ~3% of those still susceptible becoming infected". I am very sorry but this figure is not a measure of transmission. No quantitative epidemiological measures of transmission have been presented here.

line 519: "The low incidence and minimal herd immunity levels less than three years after the CHIKV introduction place 85% of the population at risk for new outbreaks" Do the authors have any evidence of that/ In which population? Herd immunity is strictly related to R0 and disease dynamics - it looks to me that the authors speculate on issues without evidence.

**Conclusions**

-Are the conclusions supported by the data presented?

-Are the limitations of analysis clearly described?

-Do the authors discuss how these data can be helpful to advance our understanding of the topic under study?

-Is public health relevance addressed?

Reviewer #1: Authors have addressed the reviewers' comments adequately and the conclusion supported by the data presented.

Reviewer #2: I rest my case regarding the title - below is the conclusion of the authors. The title should reflect what's been done, not what needs to be done.

"Additional investigations are needed to elucidate the factors driving CHIKV spread dynamics, including understanding differences with respect to dengue and Zika viruses, in order to guide prevention and control strategies for coping with future outbreaks."

**Editorial and Data Presentation Modifications?**

Reviewer #1: None

Reviewer #2: (No Response)

**Summary and General Comments**

Reviewer #1: A much improved manuscript and of interest to the readers of PLOS NTD

Reviewer #2: The paper is of interest but it still needs revision. I am happy to read it again - but there is too much from too little data and it seems that the incidence cannot be robustly estimated. There are also lots of speculative comments without references or supported by the data in the Discussion.

PLOS authors have the option to publish the peer review history of their article (what does this mean?). If published, this will include your full peer review and any attached files.

Reviewer #1: Yes: Timothy P Endy

Reviewer #2: Yes: Irina Chis Ster

Figure Files:

Data Requirements:

Reproducibility:

References

---

## [Editor Report · Decision Letter 2]

14 Dec 2023

Dear Dr.. Ribeiro,

We are pleased to inform you that your manuscript 'Dynamics of chikungunya virus transmission in the first year after its introduction in Brazil: A cohort study in an urban community' has been provisionally accepted for publication in PLOS Neglected Tropical Diseases.

Best regards,

Brett M. Forshey

Academic Editor

Abdallah Samy

Section Editor

---

## [Editor Report · Acceptance letter]

21 Dec 2023

Dear Dr.. Ribeiro,

We are delighted to inform you that your manuscript, "Dynamics of chikungunya virus transmission in the first year after its introduction in Brazil: A cohort study in an urban community," has been formally accepted for publication in PLOS Neglected Tropical Diseases.

Best regards,

Shaden Kamhawi

co-Editor-in-Chief

Paul Brindley

co-Editor-in-Chief
